# Networks of care for optimizing Primary Health Care Service Delivery in Ethiopia: Enhancing relational linkages and care coordination

Gizachew Tadele Tiruneh[1]*, Nebreed Fesseha[1], Temesgen Ayehu[2], Tamar Chitashvili[3], Mesele Damte Argaw[2], Biruhtesfa Bekele Shiferaw[1], Mikias Teferi[2], Agumasie Semahegn[2], Biruk Bogale[1], Yibeltal Kifle[4], Hillina Tadesse[1], Chala Tesfaye[1], Dessalew Emaway[1]

**1** JSI, Addis Ababa, Ethiopia, **2** Amref Health Africa, Addis Ababa, Ethiopia, **3** JSI, Washington, DC, United States of America, **4** MERQ Consultancy PLC, Addis Ababa, Ethiopia

* gizachew_tadele@et.jsi.com

**Data Availability Statement:** The dataset used and analyzed during this study is included as supplementary information to this article (S2 File).

## Abstract

### Introduction

Ethiopia has made notable progress in reducing maternal and perinatal mortality, yet challenges remain in meeting the 2030 Sustainable Development Goals. Persistent issues such as low service utilization, coupled with poor quality, fragmented care, and ineffective referral systems hinder progress. The *"Improve Primary Health Care Service Delivery (IPHCSD)"* project, implemented by JSI and Amref Health Africa since April 2022, seeks to address these gaps through a Networks of Care (NoCs) approach. This paper describes the lessons learned from implementing the NoCs approach to optimize primary health care in Ethiopia.

### Methods

The project incorporates embedded implementation science, guided by the RE-AIM (Reach, Effectiveness, Adoption, Implementation, and Maintenance) framework. Key implementation strategies co-designed included strengthening community engagement, establishing NoCs, and introducing quality improvement initiatives using the Model for Improvement. Routine program monitoring data, NoCs process evaluation, and facility service statistics were utilized for this study. Service statistics were analyzed using Student's t-test and interrupted time-series analysis to compare maternal and child care outcomes before and after the NoCs intervention, with counterfactual estimates generated to assess the intervention's impact. Qualitative data from key informant interviews were transcribed, coded, and analyzed to identify themes and patterns using Atlas.ti.

### Results

The NoCs approach has significantly enhanced relational linkages between primary health care facilities and health care providers, fostering stronger collaboration and

**Funding:** The project is funded by the Bill & Melinda Gates Foundation and implemented through a collaborative consortium of Amref Health Africa and JSI. The funder has no role in the interpretation and implications of the content in this paper. This responsibility lies solely with the authors.

**Competing interests:** The authors have declared that no competing interests exist.

**Abbreviations:** ANC, antenatal care; HEP, Health Extension Program; HEW, Health Extension Worker; IPHCSD, Improve Primary Health Care Service Delivery; IRB, Institutional Review Board; NoCs, Networks of Care; PHC, primary health care; PHCU, primary health care unit; RMNCH, reproductive, maternal, newborn, and child health; RE-AIM, Reach, Effectiveness, Adoption, Implementation, and Maintenance; VHL, Village Health Leader.

communication. This has fostered trust, improved care coordination, optimized primary health care performance, and increased health service utilization within woreda health systems. The interrupted time series analysis indicated that the rate of ANC 8+ visits was 29.8% per month higher than expected without the NoCs strategy (Coef: 2.39; p-value < 0.01) and an 18.4% increase in obstetric complications managed (Coef: 1.71; p-value = 0.050), with a 43% overall increase. Perinatal mortality decreased by 34%, from 31.3 to 20.1 per 1,000 births [t-test: 2.12; p-value: 0.040)].

## Conclusion

The NoCs approach in Ethiopia has proven effective in enhancing the relational elements, care coordination, and quality of primary health care services, leading to better maternal and child health outcomes. The findings expand the existing body of research on NoCs implementation best practices and further confirm that it provides a scalable model for strengthening health services in low-resource settings.

## Introduction

Ethiopia has made remarkable progress in reducing maternal and child mortality rates [1, 2] through the primary health care (PHC)/Health Extension Program (HEP) [3–5]. However, these achievements alone may not be sufficient to meet the Sustainable Development Goals 3 by 2030. Several challenges impede further progress, including sub-optimal quality of care [6], fragmented [7, 8] and uncoordinated health care [9], and poorly functioning referral systems [10] within primary health care facilities. Addressing these gaps necessitates collaborative efforts between PHC facilities and communities.

One promising strategy to tackle these challenges is the implementation of Networks of Care (NoCs). Networks of care is a collection of health facilities and health workers purposefully interconnected to promote relational linkages, teamwork, and collaborative learning to provide equitable integrated, comprehensive, patient-centered, and coordinated health services throughout the lifecycle across all levels of health service delivery for women and children [11, 12]. This approach has improved maternal and child health outcomes in various countries including Madagascar, Nepal, Nigeria, the Philippines, and Tanzania [12–14]. Previous case studies demonstrate that networks of health facilities, characterized by clear agreements, supportive linkages, feedback loops, standards-based clinical skill-building, mentoring with routine case reviews, and joint learning forums, significantly enhance quality care, service delivery, and maternal and child health outcomes [12, 14, 15].

Since April 2022, JSI and Amref Health Africa have been implementing the *"Improve Primary Health Care Service Delivery (IPHCSD)"* project in Ethiopia to strengthen the functionality and bidirectional linkage across PHC delivery platforms using the woreda (i.e., district)-wide NoCs approach. This involves designing and implementing strategies that connect public and private PHC facilities and communities through administrative and clinical support mechanisms, and implementing quality improvement activities to enhance relational linkages among health workers and facilities, care coordination, strengthen referral linkages, and ultimately improve the delivery of reproductive, maternal, newborn, and child health (RMNCH) services and care outcomes at the woreda level.

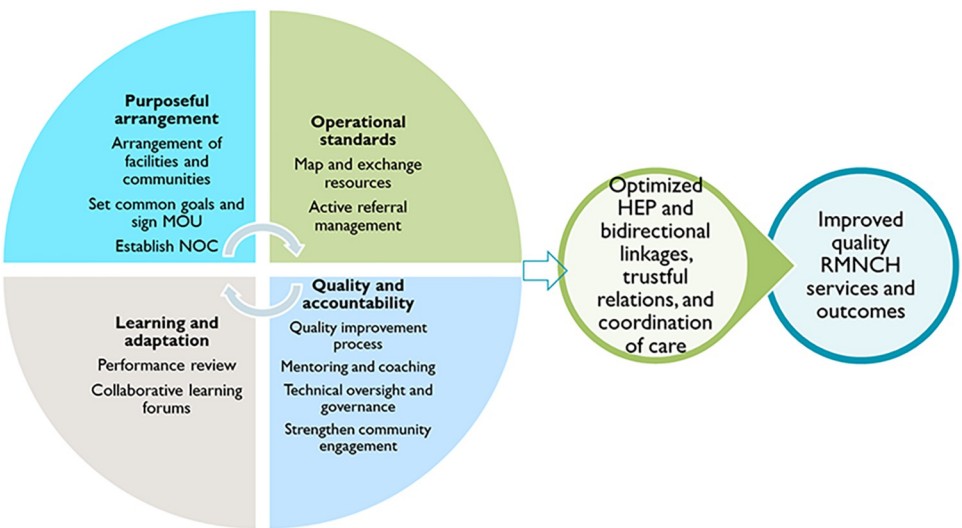

**Fig 1. Conceptual framework depicting the relationship between NoCs domains, bidirectional referral linkages, and RMNCH outcomes.**

Networks of care serve as a foundational platform with shared governance for collaborative learning crucial for effective quality improvement and community engagement initiatives. By engaging communities and improving quality processes, NoCs collaboratively identify gaps and inefficiencies and address them through innovative local solutions. Accordingly, the conceptual framework outlined in Fig 1 presents the NoCs approach to optimizing bidirectional linkages, fostering trustful relationships, and enhancing care coordination, ultimately improving the performance of PHC and optimizing the HEP to enhance RMNCH outcomes. It is structured around four key elements: purposeful arrangement, operational standards, quality and accountability, and learning and adaptation [14].

This paper describes the lessons learned from implementing the NoCs approach within primary care settings, alongside quality improvement and community engagement strategies, in the agrarian and pastoral communities in Ethiopia.

## Methods

### Context

The Ethiopian health system is organized into primary-, secondary-, and tertiary levels of care to offer comprehensive health services for citizens across the country. The primary level of care comprises primary health care units (i.e., health centers and health posts) and primary hospitals. At the grassroots level, health posts are situated in rural areas and staffed by Health Extension Workers (HEW) who deliver basic preventive and promotive services. Health centers provide a broader range of outpatient and maternal health services, while primary hospitals act as referral points for inpatient care, surgeries, and advanced medical treatments. General and specialized hospitals provide specialized secondary and tertiary-level services, respectively.

The Ministry of Health introduced the HEP Optimization Roadmap (2020–2035) in 2020 to enhance access to and quality of essential health service delivery [16]. As a key initiative, the roadmap aims to restructure health service delivery by mapping health posts in relation to health centers. This involves converting the farthest health posts into comprehensive ones capable of providing comprehensive essential health services and merging health posts located

very close to health centers or primary hospitals. Other strategic initiatives in the roadmap for optimizing HEP include reconfiguring community volunteers to enhance community engagement and interaction with the PHC system.

## Project description

Since 2022, the project has implemented targeted HEP roadmap strategic objectives in three phases: pressure test, test of scale, and creating a prototype for full-scale national implementation. From April 2022 to March 2024, the project has been pressure testing strategies in 14 woredas across nine regions, focusing on equitable access, service quality, oversight and accountability, and evidence generation for optimizing PHC service delivery. From April 2024 to March 2027, we will expand to 30 additional woredas to assess national adaptation feasibility, building on lessons learned to optimize PHC service delivery across the regions.

Using the NoCs approach, we aimed to strengthen PHC and achieve universal health coverage by enhancing access to comprehensive RMNCH services. The NoCs was designed with four key domains (S1 Fig):

1. **Purposeful Arrangement:** We connected service delivery points and communities within each woreda with shared governance. In September 2021, we mapped service sites, establishing the NoCs in all 14 woredas, which included six primary hospitals, 66 health centers, 15 comprehensive health posts, 254 basic health posts, and 53 private PHC facilities. The network functions and responsibilities (S1 Table) within the NoCs involve a coordinated effort across community/ health posts, health centers/clinics, primary hospitals, and woreda health offices to enhance relational linkages, care coordination, and resource sharing ultimately improve maternal and child care outcomes The arrangement of facilities in the woreda-wide NoCs is presented in S2 Fig. In addition, in pastoral regions, the NoCs platform supplemented by the hub-and-spoke model to support basic health posts far from health centers, through the nearest comprehensive health posts (hub) with better capacity, improving functionality, referral linkage, feedback, and initiating new services at basic health posts (spoke).

   Between October and December 2021, co-design workshops involving 957 participants from various health sectors established shared goals and operational standards for the NoCs. Key priorities included increasing skilled birth attendance by 20–30%, improving compliance with RMNCH care recommendations, and reducing perinatal mortality by 30%. The workshops culminated in a Memorandum of Understanding formalizing the NoCs's establishment and guiding collective efforts to enhance RMNCH outcomes.

2. **Operational Standards:** We adapted evidence-based standards and tools to enhance PHC performance and RMNCH care. Facilities implemented service standards, improved communication and referral systems, and engaged in quality improvement processes, focusing on providing integrated, person-centered care.

3. **Quality and Accountability:** We improved clinical care quality through coordinated efforts, adherence to standards, and continuous quality improvement using Plan-Do-Study-Act cycles. Community engagement was emphasized through Village Health Leaders (VHLs) and Women Development Army groups, which mobilized the community.

4. **Learning and Adaptation:** We maintained effective NoCs functions through coordinating committees, learning sessions, and performance evaluations. This facilitated collaborative learning and data-driven adaptations, ensuring sustainability

## Embedded implementation science research

The project employed embedded implementation science research to engage relevant stake-holders in the design, implementation, and evaluation phases. The processes, adaptations, and effectiveness of implementation strategies beyond their immediate outcomes were monitored to inform the high-fidelity implementation of HEP roadmap interventions to enhance access to and utilization of RMNCH services. To achieve this, the RE-AIM framework (Reach, Effectiveness, Adoption, Implementation, and Maintenance) [17] was utilized to assess the outcomes of our implementation strategies. Additionally, the Expert Recommendations for Implementing Change protocol [4] were employed to engage program managers in systematic planning. This involved contextualizing and validating preliminary strategies, and selecting priority approaches to operationalize targeted and innovative PHC service delivery modalities under the HEP framework. Key implementation strategies co-designed included strengthening community engagement, establishing NoCs, and introducing quality improvement initiatives using the Model for Improvement. Stakeholders were continuously engaged throughout the project complementation and evaluation to drive change and adopt innovations, fostering a culture of co-production of knowledge, reflection, and collaborative learning. This approach ensures timely feedback provision for implementation, enhances ownership of research findings, and facilitates seamless integration of strategies into routine health system workflows. Adaptive strategies were documented using the Framework for Reporting Adaptations and Modifications to Evidence-based Implementation Strategies [18].

## Data

For this study, we utilized routine program monitoring data collected during the project implementation period (April 2022 to March 2024), insights from the NoCs learning documentation, and facility service statistics. The NoCs learning documentation (process evaluation) was conducted from 28-03-2024 to 25-04-2024 and included 11 comprehensive health posts, 66 health centers, and 14 associated hospitals. Service statistics were gathered from October 2021 to March 2024. Additionally, a programmatic qualitative study was conducted using key informant interviews with 16 health care providers, program managers, and/or NoCs coordination committees at intervention woredas.

Qualitative data were collected in March 2024 by the project Monitoring and Evaluation and program managers using an interview guide with open-ended and probing questions to gather insights from program implementers. The main topics of discussion included the differences between NoCs and existing linkages, participatory and contextualized design, roles and responsibilities of network members, care coordination and communication, operational impact on PHC and HEP, collaboration and oversight, adherence to standards and quality improvement, monitoring and evaluation mechanisms, support systems and resource sharing, barriers and facilitators, unintended consequences, and areas for improvement (S1 File). Interviews were conducted at convenient locations within the participants' work environment.

## Measurement

Table 1 below presents the key indicators measured, along with their definitions and data sources.

## Analysis

Service statistics were extracted from facility records to monitor trends in RMNCH outcomes. Student's t-test was used to compare the mean perinatal mortality rate, ANC 8+ visits, and

**Table 1. Matrix of indicators and data sources for implementation of NoCs in primary care facilities in Ethiopia, 2024.**

| RE-AIM domains | Indicators measured | Definitions and Calculations | Data sources |
|---|---|---|---|
| **Reach**<br><br>*The degree to which an intervention-eligible population receives it (Coverage of RMNCH services)* | Antenatal care (ANC) 8+ contacts coverage | The percentage of pregnant women who have 8 or more ANC contacts. It is calculated as the number of ANC contacts (8 or more), divided by the expected number of pregnancies in the NoCs catchment area, and multiplied by 100. The expected number of pregnancies was estimated based on 3.2% of the live births within the catchment population. | Service statistics, facility records |
| | Proportion of obstetric complications managed | The percentage of women with direct obstetric complications (including abortion complications, postpartum hemorrhage, obstructed or prolonged labor, and puerperal sepsis) who were treated at health centers and their corresponding hospitals. It is calculated as the number of women with direct obstetric complications treated in health centers and hospitals, divided by the estimated number of women expected to experience obstetric complications (i.e., 15% of expected births), [19] and multiplied by 100. | |
| | Early ANC coverage | The proportion of 1st ANC contacts within 1st trimester. It is calculated as the number of ANC visits within the first trimester, divided by the expected number of pregnancies in the NoCs catchment area, and multiplied by 100. The expected number of pregnancies was estimated based on 3.2% of the live births within the catchment population | |
| **Effectiveness** *The impact of NoCs strategy on perinatal mortality reduction* | Perinatal mortality | Perinatal mortality refers to the death of a fetus after 22–28 weeks of gestation or a newborn within the first 7 days of life. Perinatal mortality was calculated using an estimated annual birth rate of 3.4% and a pooled perinatal mortality rate of 51.3 per 1,000 births, as reported in a previous systematic review in Ethiopia [20]. | Service statistics, facility records |
| **Adoption**<br><br>*Uptake of NoCs model* | NoCs established | Number of primary health care facilities established NoCs | Service statistics, Project MIS |
| | Referral protocol adherence | Proportion of maternal referrals that adhere to specific protocols (i.e., referral slip, advance call, and feedback) | |
| | Enhanced bidirectional linkage | Enhanced bidirectional linkage refers to a system that strengthens communication, trust, referral processes, and feedback loops between NoCs facilities and communities, facilitating resource sharing, collaborative problem-solving, and adherence to clinical and referral standards across a network of primary and community health services. | |
| **Implementation**<br><br>*Degree of implementation of NoCs strategy, functionality and maturity, and contextual factors that hinder or facilitate the delivery of the implementation strategies* | Fidelity: NoCs functionality and maturity | Fidelity is the degree to which the NoCs is implemented as designed, aligning with its four domains: Purposeful Arrangement, Operational Standards, Quality and Accountability, and Learning and Adaptation. Fidelity was measured by evaluating adherence to critical components and core functions using functionality and maturity metrics, while also considering ongoing implementation challenges. Specifically, **NoCs functionality** assesses how effectively the network operates to achieve its shared goals. This includes key measurement parameters such as establishment and governance, community engagement, bidirectional referral, resource sharing, implementation of clinical standards and reforms, quality improvement, capacity enhancement, learning, gender integration, respectful, people-centered care, engagement of private facilities, and bidirectional accountability. While **NoCs maturity (woreda level)** describes the development stages of the NoCs across four domains: 1) purposeful arrangement, 2) operational standards, 3) quality and accountability, and 4) learning and adaptation. Each domain included sub-domains or key activities that were evaluated on a five-point scale. A score of 1 indicated a non-existent, emerging, or ad hoc stage (least mature), 2 indicated a repeatable, introduced, or established stage, 3 indicated a defined stage, 4 indicated a managed stage, and 5 indicated an optimized stage (highest maturity). | Project MIS, qualitative interviews |
| | Support system | Number of NoCs coordinating committee meetings, joint performance reviews, and learning forums held to share best practices and experiences. | |
| | Contextual challenges and adaptions | Contextual factors that hinder or facilitate the delivery of the implementation strategies | |
| **Maintenance**<br><br>*Extent of NoCs maintained or institutionalized in a PHC setting (Sustainability)* | Optimized HEP and improved performance of the PHC system | Optimized PHC service delivery is a coordinated service delivery that strengthens networks, improves health worker skills, mobilizes resources, and fosters community ownership to enhance access, quality, and efficiency in primary healthcare services. | Qualitative interviews |

obstetric complications managed before and after the intervention. An interrupted time-series analysis [21] was conducted to assess the rate of ANC 8+ visits and the management of obstetric complications across the NoCs. The pre-intervention period spanned from October 2021 to September 2022, while the intervention period covered October 2022 to March 2024. Time series models, adjusted using the Cumby-Huizinga test to account for lags in autocorrelation, were fitted to evaluate differences between the pre-intervention and intervention periods in the rate of ANC 8+ visits and complications managed. Counterfactual estimates, representing what would have occurred without the intervention, were generated using the 'itsa' command in Stata. The impact of the NoCs strategy was estimated using the 'lincom' Stata post-estimation command.

All interviews were audio-taped with the consent of the study participants and the records were transcribed verbatim by the project Monitoring and Evaluation team. The qualitative data from key informants were transcribed, coded, and thematically analyzed using Atlas.ti. Priori codes were generated from the interview guide. Reports of quotations, themes, and patterns of interrelationships between the themes were identified and quantitative data were integrated during both the presentation and interpretation stages.

## Ethics

The data used in this study was primarily collected for program quality improvement and monitoring. For this study, we conducted a secondary analysis of the data originally collected for quality improvement purposes. The project also obtained ethical review and clearance for the implementation research from the Ethiopian Public Health Association Institutional Review Board (IRB) on February 19, 2024, with reference number EPHA/OG/159/24.

For the retrospective service statistics from medical records, data were collected in monthly aggregates and fully anonymized prior to access. The IRB waived the requirement for client informed consent; however, we obtained consent from facility directors. Informed written consent was also secured from key informants for qualitative interviews. All information obtained from key informants was de-identified before analysis, and we maintained strict confidentiality throughout all stages of the study. All interviews were conducted with informed consent and in accordance with international ethical standards.

## Results

The findings of this paper are presented according to the RE-AIM implementation outcomes 1) its *implementation fidelity* including challenges and program adaptations, 2) its effect on enhancing bidirectional linkages and improving the performance and efficiency of the PHC system (*Adoption and Maintenance*), and 3) improving quality and continuity of care (*Reach and Effectiveness*).

### Implementation fidelity, challenges, and adaptations

**Implementation fidelity.**   A sizable progress in NoCs implementation fidelity was achieved. The NoCs functionality score improved from 56% to 89% in agrarian settings and from 38% to 60% in pastoral settings (Fig 2). Improvements included enhanced coordination, standardized referral practices, and increased resource sharing among facilities. Coaching and mentorship strengthened at primary care levels, improving efficiency, access, and quality of care.

In March 2024, the NoCs maturity matrix was evaluated using the four domains: purposeful arrangement, operational standards, quality and accountability, and learning and adaptation. The maturity levels ranged from Ad-Hoc to Established, with agrarian settings showing higher

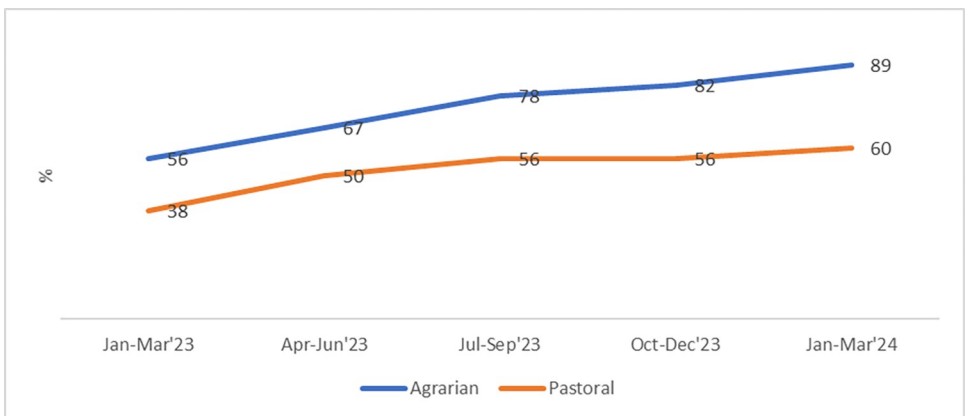

**Fig 2. Trends in NoCs functionality score, January 2023 to January 2024.**

maturity levels than pastoral settings. It highlights that purposeful arrangement and quality and accountability domains showcase higher maturity levels, while the learning and adaptation domain is at the comparatively lowest maturity level, characterized by an emergent stage highlighting the need to enhance data use (Fig 3).

**Support system.** Regular NoCs coordinating committee meetings, joint performance reviews, and learning forums were held to share best practices and monitor client experiences. The project organized two rounds of national-level performance reviews focused on learning and setting future goals. During these review forums, participants discussed the need to modify and adapt strategies to address implementation challenges and to ensure a more integrated and cohesive support system within the health care framework.

**Implementation challenges and program adaptations.** The project faced a range of challenges, including civil conflict, outbreaks of malaria, cholera, and measles, flooding, and competing health system priorities. Additionally, high staff turnover, dispersed community settlements, frequent mobility due to drought and conflict, budget shortages, and inaccessibility to health facilities due to long distances were significant obstacles to ensuring access to quality health care through NoCs in the pastoral settings of Ethiopia. One of the key

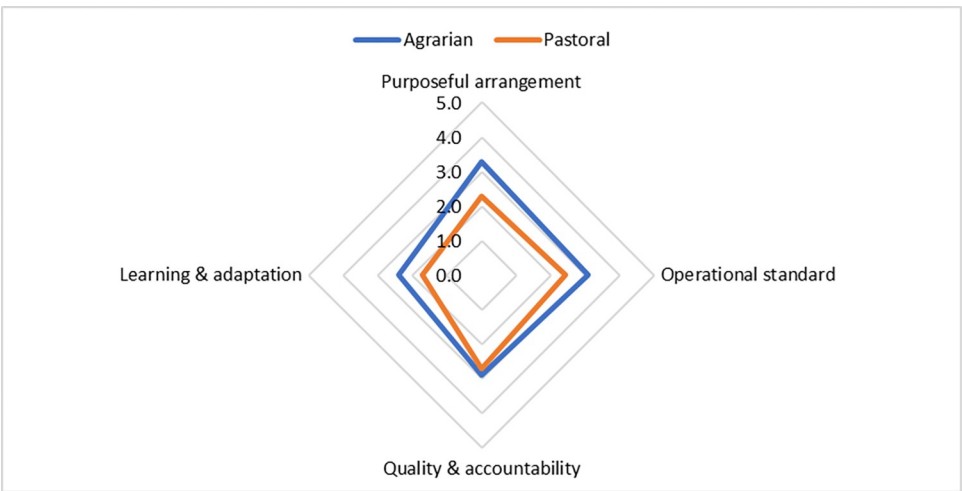

**Fig 3. NoCs maturity score by domains, October 2023 to March 2024.**

informants reported the complex nature of barriers to the effective implementation of the NoCs strategy, highlighting the need for system improvements:

> *"Implementation of NoCs demands structural arrangements, which require higher-level commitment, decision-making, and budget allocation to facilitate technical assistance among the catchment facilities."*—**Health Center Director, pastoral**

During implementation, it became evident that the broad goals of NoCs lacked clarity and alignment of objectives across all levels, along with limited engagement with private health care facilities. To address these issues and streamline NoCs implementation for clarity and effectiveness, the project specified precise objectives for NoCs and defined activities under each domain across all facilities and networks. Efforts were made to optimize engagement with private health care facilities through tailored strategies and partnerships aimed at broadening participation and improving health care delivery. Additionally, the project facilitated the integration of NoCs collaborative learnings into regular PHCU-level meetings and catchment-based mentorship strategies to foster continuous improvement, share best practices, and effectively address implementation challenges. Close collaboration with the health system and local administration played a crucial role in overcoming these challenges, enabling effective crisis responses and advancing the operationalization of strategies in the intervention woredas.

## Enhanced bidirectional linkages

The NoCs has played a crucial role in connecting the health system, community, and beyond to address the common goal of improved access to quality health care. Implementing NoCs has enhanced trust, communication, referral systems, feedback loops, and resource sharing, substantially supporting health facilities in assisting each other. It has facilitated the establishment of new comprehensive health posts, enhanced health care provider competencies, and ensured better adherence to clinical standards. Additionally, the hub-and-spoke model has improved health care access and coordination in pastoral settings. For example, in Awash Fentale woreda of Afar and Dire woreda of the Oromia region, the hub-and-spoke model ensures basic health posts receive support from nearby Doho and Madacho comprehensive health posts, respectively.

One of the key informants from the primary health care unit has remarked that "*. . .NoCs is a system that connects our health facilities and with the community to work together. . .*"— **Health Center Director, Pastoral**

**Adherence to referral protocols.**   Referral processes improved after the NoCs intervention (Table 2); From October 2022 to March 2024, a total of 8,190 maternal referral cases were made. Of these, 91% were sent with referral slips, 30% of referrals included advance communication, and 17% provided feedback.

The NoCs learning documentation conducted in March 2024 through interviews with service providers and program managers at intervention woredas revealed the following advancements:

**Table 2. Proportion of maternal referrals that adhere to specific protocols, October 2021-March 2024.**

| Maternal referrals | Oct'21-Sep'22 | Oct'22-Mar'24 |
|---|---|---|
| Advance call | 1,089 (22%) | 2,486 (30%) |
| Referral slip | 3,942 (79%) | 7,456 (91%) |
| Feedback provided | 748 (15%) | 1,415 (17%) |

**Strengthened bidirectional referral.** Key informants affirm NoCs's role in facilitating connections and referrals throughout the woreda. According to the health center HEP coordinator, there were previously disjointed arrangements and referral linkages among health posts, health centers, and hospitals. However, there has been a notable transformation *"now the linkage among health post and health centers and even with private health facilities strengthened."* Before the NoCs coordination platform establishment and training, referrals often lacked formal arrangements and communication with the subsequent referral destination, relying solely on oral communication. Since the implementation of the NoCs, *"the referral process has been through standard referral formats and advanced communication to the referred facilities."*

Another key informant noted, *"Before NoCs, there was no consistent provision of feedback from hospitals to health centers or from health centers to health posts. However, now we are communicating very frequently. There are also visible and documented referral papers and feedback at all networked health facilities. We follow up with our referred patients and clients by phone call."*—**Health care provider, pastoral**

**Improved coordination and communication.** The NoCs platform is evident through regular meetings chaired by the woreda administrator, fostering discussions and solutions to enhance RMNCH services. NoCs mentorship promotes familiarity among staff from different health facilities. Additionally, a shared NoCs Telegram page facilitates the exchange of information, resources, updates, and challenges among member facilities.

*"Improvement has been observed in reducing client complaints. We have a liaison officer who communicates with other catchment health facilities via phone, Telegram, etc."*—**Primary Hospital Director, Pastoral**

Furthermore, the routine health data of the Ethiopian Health Center Reform Implementation Guidelines implementation, health center-health post linkage score improved from 77% to 93% between September 2022 and February 2024 (S3 Fig). This progress is likely to be attributed to the implementation of the NoCs approach.

## Optimized HEP and improved performance and efficiency of the PHC system

**Strengthened community engagement.** The NoCs establishment has involved various stakeholders including kebele-level structures, religious leaders, HEWs, traditional leaders, village health leaders, and community council members. The stakeholder representatives were closely working and participating in the implementation of NoCs and supporting the health system through social mobilization and awareness creation activities.

Under the PHCU network, health center providers coordinate HEWs and ensure they perform their tasks correctly. While at the community level network, HEWs gather VHLs at the health post to discuss community health service provision. VHLs then engage with the community and hold discussions. The project facilitated the optimization of Women Development Army and VHL strategies after contextualizing for agrarian and pastoral communities. In agrarian settings, the project supported the training and deployment of 2,484 VHLs, averaging 13 VHLs per kebele, to promote the health of 80 to 100 households within their catchment areas and support HEWs. In pastoral settings, the VHL approach was adapted, resulting in the recruitment of 1,306 VHLs. Across all intervention woredas, VHLs played a pivotal role in facilitating health literacy activities in targeted households, identified pregnant mothers, newborns, defaulters, zero-dose children, malaria prevention, outbreak rumor identification, vaccination campaigns, and strengthened links with the health system.

The NoCs arrangement has fostered enhanced community ownership, as seen in the active involvement of community members in various initiatives. Contributions such as funds, labor, and materials for the rehabilitation of health facilities demonstrate a strong sense of community responsibility. Additionally, VHLs are effectively identifying pregnant and postnatal mothers in need of services, with support from local community structures like elders, religious leaders, and traditional arrangements, facilitating education and referrals to health facilities.

*"Thanks to the NoCs arrangement, a strong sense of community ownership has emerged. For example, community members contributed funds, labor, and wooden materials to renovate the basic health post, previously nonfunctional."*—**Vice Head, Woreda Health Office, Agrarian**

**Improved HEP performance.** The implementation of the woreda-wide NoCs strengthened the linkages between health centers and health posts, as well as enhanced team-based outreach services and support for HEWs. We have learned that since the NoCs's introduction, the connection between health centers and health posts has improved, leading to better HEP performance.

*"The NoCs arrangement has improved our community-level activities. As a result, we observed improved performance of HEP."*—**HEP Coordinator, Agrarian**

This is because community health services are now delivered collaboratively by health care providers and HEWs, allowing HEWs to enhance their skills and gain more experience.

*"Previously, the integration of health post and health center was weak, but currently, there is a big improvement in technical support in terms of frequency of support and content of the supervision that improved our skills."*—**HEW, Agrarian**

**Enhanced knowledge and skill transfer.** Within this network, dedicated efforts were directed toward enhancing the skills of health center staff and HEWs through supportive supervision, mentorship, and coaching. Hospital mentors offer guidance to health center staff, who in turn provide quarterly mentorship and coaching sessions for HEWs, focusing on specific RMNCH skill gaps.

*"In the NoCs platform, our ongoing efforts revolve around enhancing the capabilities of health center staff and HEWs through supportive supervision, mentorship, and coaching. One notable initiative involves conducting clinical mentorship sessions to refine providers' skills in child delivery and abortion care at health centers and comprehensive health posts. These sessions serve as a platform for the exchange of valuable knowledge and skills among health care professionals."*—**Quality Coordinator, Primary Hospital, Agrarian**

The performance of HEWs on community-based outreach and home visit services has improved due to improved skills, motivation, and close monitoring and accountability.

*"When they [HEWs] went outreach, the service they provide to the people is improving too. Because, they are improving in terms of knowledge, and it is convenient for monitoring."*—**Health Center Director, Agrarian**

**Enhanced resource mobilization and sharing.**   The NoCs facilitates efficient resource sharing among health care facilities, ensuring essential supplies reach where they are needed most. Through NoCs coordination, facilities receive support in terms of equipment and supplies, enabling smooth operation and service expansion. The NoCs interventions also found to be instrumental to improve the access to medicines and equipment, and enhance the functionality of health posts.

> *"Since we don't have a separate budget for supply and equipment purchase to the* comprehensive health post*, the NoCs arrangement allows us to get medicines, equipment, and supplies from the primary hospital and the health center to operationalize our facility. They are continually providing us with the necessary medicines and equipment through our agreement with the NoCs."*—**Comprehensive Health Post Coordinator, Agrarian**

This approach has led to notable improvements, including mobilizing crucial resources to establish new comprehensive health posts and enhance collaborative learning among facilities.

> "This initiative [NoCs], coupled with continuous support from IPHCSD project, facilitated the provision of essential supplies and medical equipment to the comprehensive health post from networked health facilities."—**Woreda Health Office MCH director, Pastoral**

## Improved utilization, quality and continuity of care

The NoCs initiative has improved continuity of care by ensuring seamless referral of clients from lower-level facilities to higher-level ones, resulting in optimized maternal and child health service delivery. Regular mentorship sessions have strengthened hospital and health center support. In addition, community-level activities facilitated by VHLs have increased client flow to health centers, particularly for ANC and delivery services. This has also improved early ANC booking and vaccination coverage within health center catchment areas.

> *"The NoCs arrangement has strengthened our community-level activities with Village Health Leaders, increasing client flow to the health center. As a result, more pregnant mothers are now seeking ANC and delivery services."*—**HEP Coordinator, Agrarian**

Facility service statistics demonstrated significant improvements in the maternal and child health continuum of care. For instance, ANC 8+ visits increased from 2% to 13% in agrarian settings and from 1% to 7% in pastoral settings. Interrupted time series analysis indicated that the rate of ANC 8+ visits was 29.8% per month higher than would have been expected without the NoCs strategy (Coef: 2.39; p-value < 0.01) (Fig 4).

Similarly, the proportion of pregnant women receiving ANC visits in the first trimester showed a notable increase, rising from 9% to 21% in agrarian settings and from 7% to 17% in pastoral settings (Fig 5).

The NoCs strategy, which focused on strengthening the skills and competencies of health workers through enhanced training, coaching, and mentorship, likely contributed to the increased application of these skills in clinical practice. As a result, the management of obstetric complications significantly improved from 38% in the pre-intervention period (October 2021–September 2022) to 52% post-intervention (October 2022–March 2024) [t-test: 3.48; p-value < 0.01]. Interrupted time series analysis revealed that, a year after the intervention, the rate of obstetric complications managed was 18.4% higher per month than expected without

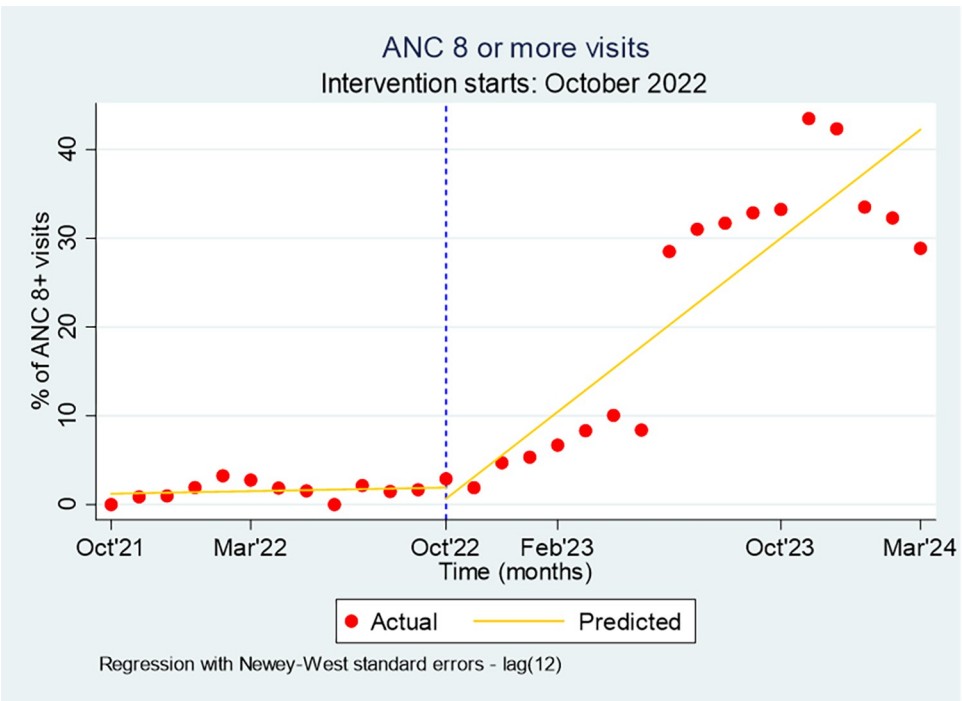

**Fig 4. Percentage of pregnant women with ANC 8+ visits, October 2021-March 2024.**

the NoCs strategy (Coef: 1.71; p-value = 0.050). This equates to a 43% relative increase, calculated as the absolute change divided by the counterfactual (Fig 6).

Furthermore, the combined community-facility quality improvement strategy led to a 34% reduction in perinatal mortality rates (t-test: 2.12; p-value: 0.040). The time series model also indicated that, a year after the intervention, there was a 14.8% monthly absolute decline in

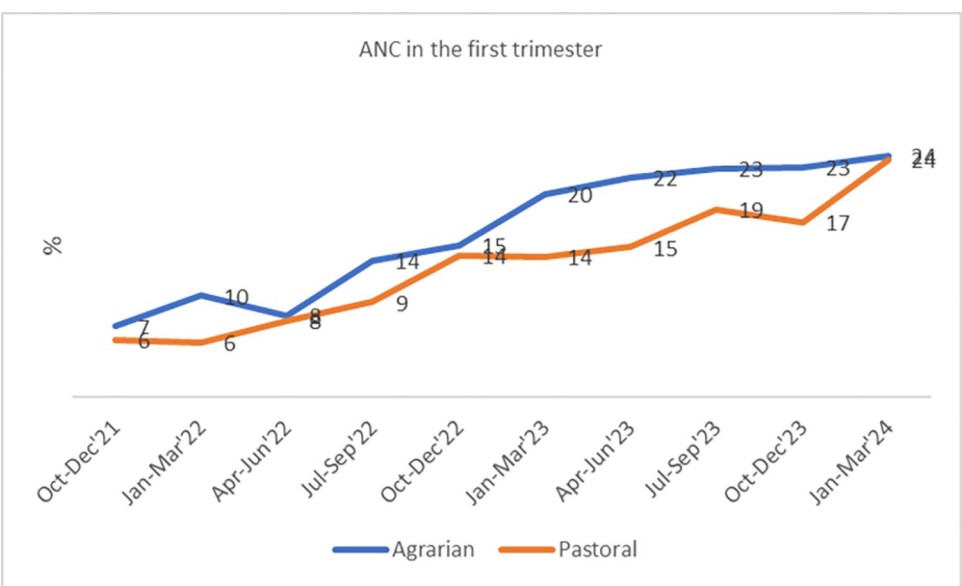

**Fig 5. Trends in coverage of early ANC visits, October 2021-March 2024.**

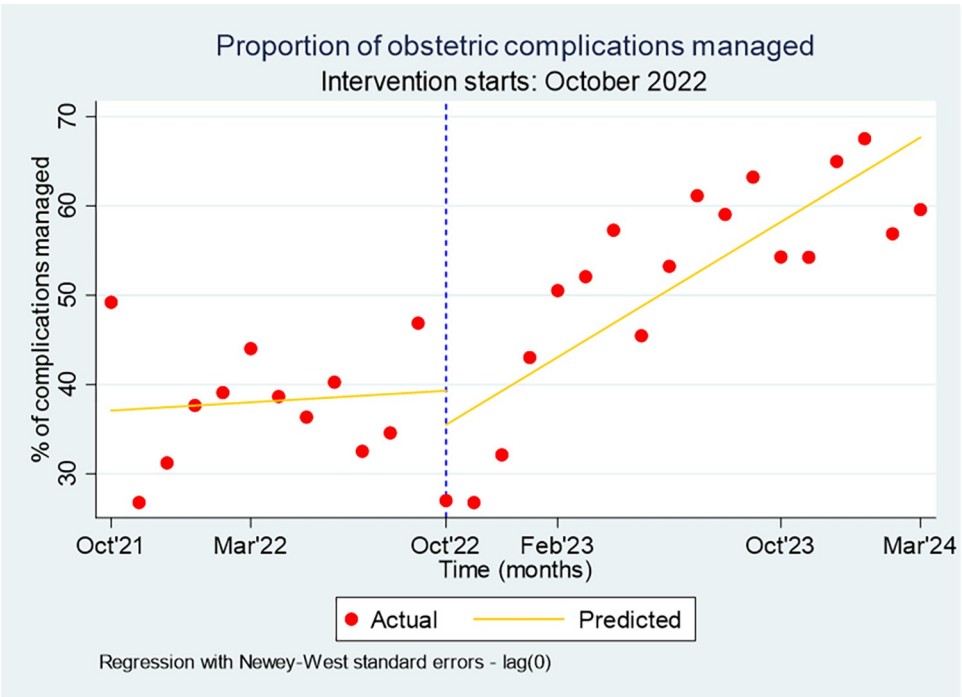

**Fig 6. Proportion of obstetric complications managed in intervention woredas, October 2021-March 2024.**

perinatal mortality, which exceeded expectations without the NoCs strategy, resulting in a 95% relative reduction (absolute change divided by the counterfactual). As depicted in the figure below, perinatal mortality significantly decreased from an average of 31.3 per 1,000 births from March 2022 to February 2023 to 20.1 per 1,000 births from March 2023 to April 2024. According to run chart rules, this change marks a significant shift from the established trend when compared to the baseline median (Fig 7).

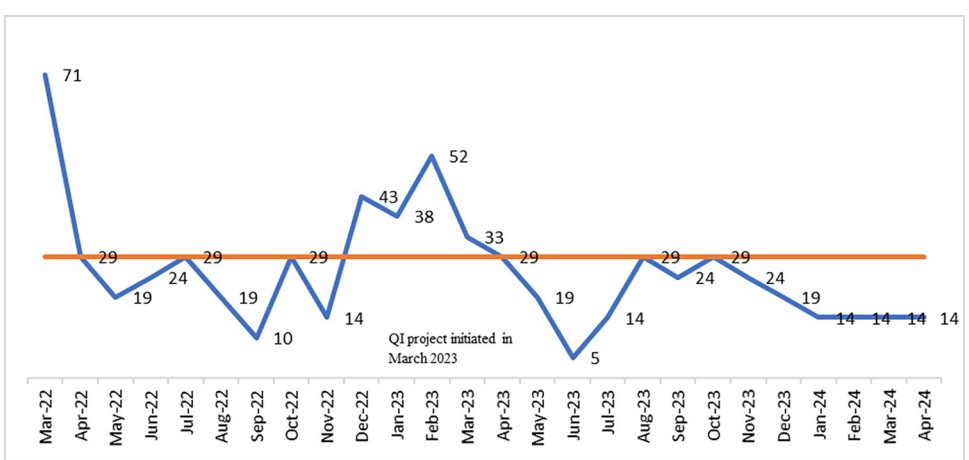

**Fig 7. Perinatal mortality rate per 1000 births at South Bench woreda, March 2022 to April 2024.**

## Discussions

Implementing the NoCs approach integrated with quality improvement and community engagement strategies in PHC systems has highlighted the importance of strengthening both administrative and technical connections and the critical need for multi-level collaboration and robust communication channels. This approach has been instrumental in building trust, enhancing communication, and boosting efficiency among stakeholders. As a result, there has been better care coordination within woreda health systems, optimized HEP/PHC performance, and improved use of health services. The successful incorporation of quality improvement and community engagement strategies underscores the benefits of collaborative efforts among communities, health care facilities, and woreda authorities. By addressing challenges at various levels of the woreda health system, these combined efforts have improved the continuity of ANC services, increased management of obstetric complications, and significantly reduced perinatal deaths.

Networks of care serve as vital platforms for collaborative quality improvement and community engagement. Through these initiatives, gaps within network facilities are pinpointed, prompting targeted interventions and enhanced coordination. Previous studies underscore that involving communities in these networks and quality improvement efforts significantly boosts maternal health program effectiveness [12, 22, 23]. By engaging community members, local leaders, and stakeholders in service design, implementation, and evaluation, programs become more accessible, culturally sensitive, and widely utilized. Initiatives like participatory learning groups, community health committees, and training for health workers increase antenatal visits, skilled birth attendance, and postnatal care, ultimately lowering maternal and neonatal mortality rates [24–26]. These engaged networks build trust between health care providers and communities, ensuring interventions meet local needs sustainably.

NoCs focuses on creating intentional connections among health facilities and workers to strengthen health systems through collaborative learning, problem-solving, resource sharing, and accountability, ultimately leading to improved care outcomes. Previous studies also suggest NoCs would improve maternal and child health, particularly in resource-constrained settings by enhancing the coordination and continuity of care by integrating various health facilities and services, focusing on relational elements such as teamwork, trust, and communication [14, 15, 27].

Networks of care are essential for enhancing relational linkages, care coordination and quality in health care settings due to their ability to foster integration among diverse health care providers and facilities. By creating interconnected systems that facilitate communication, collaboration, and resource-sharing, networks of care ensure that patients receive seamless and comprehensive care across different points of service [28, 29]. This integration is particularly crucial in addressing complex health needs and chronic conditions, where multidisciplinary approaches are required for effective management [30]. Furthermore, networks of care support the implementation of standardized protocols and evidence-based practices, which contribute to improved clinical outcomes and patient safety [31]. They also enable health care organizations to leverage collective expertise and resources, leading to more efficient use of health care resources and reduced duplication of services [32]. Overall, networks of care promote a patient-centered approach to health care delivery by enhancing continuity of care, optimizing care transitions, and ultimately improving the overall quality of health care services [11, 12, 14, 15].

This study documented that NoCs has contributed to upgrading comprehensive health posts and health centers to provide comprehensive services through effective resource sharing. These findings underscore the crucial role of NoCs in strengthening primary health care

systems, particularly in resource-limited settings. The collaborative and supportive nature of NoCs helps address challenges related to resource scarcity in primary care facilities. Previous studies indicate that NoCs are a promising solution for providing support and resources to remote and disadvantaged primary care workers and health facilities [33]. Additionally, NoCs holds promise for optimizing basic health posts and implementing the HEP optimization roadmap in Ethiopia through tailored technical and administrative support.

Despite the potential of NoCs, there is limited rigorous evidence of its effectiveness and impact [12]. Previous studies have largely focused on clinical outcomes and descriptive case studies, lacking comprehensive evaluations and implementation research [12]. This study aimed to document the lessons learned and implementation experiences of the NoCs within the rural Ethiopian PHC system. Using implementation research, this study examined mechanisms shaping networks between PHC facilities and communities. It contributes to the global literature by providing insights into how the network operates and influences PHC service delivery. Conducted as embedded implementation research, it involved continuous monitoring and adaptation of strategies to address contextual factors, ensuring both fidelity and contextual appropriateness for potential generalization. The analysis of NoCs functionality, maturity, and overall implementation fidelity contributed to the existing knowledge base, particularly by highlighting gaps and underscoring the need to enhance data use for ongoing learning and adaptation. Future research should prioritize simplifying and validating metrics for monitoring and evaluating the implementation and effectiveness of the NoCs.

Adapting programs to local contexts is crucial for ensuring strategies are both practical and effective. High-fidelity implementation improves PHC performance, health outcomes, and sustainability. However, due to the study's limited duration, certain RE-AIM measures, such as sustainability (Maintenance), were not fully assessed, limiting the evaluation of NoCs' maturity and integration into health system processes. Additionally, the absence of a comparison group may weaken the evidence from the time-series analysis, and the functionality and maturity matrix could be influenced by assessor bias and implementation variability.

## Conclusion

In conclusion, this study demonstrated that implementing the NoCs approach integrated with quality improvement and community engagement strategies in PHC systems has demonstrated the critical importance of enhancing both administrative and technical connections. This integrated approach has proven instrumental in fostering trust, improving communication, and increasing efficiency among all stakeholders involved. As a result, there has been a noticeable enhancement in care coordination within woreda health systems, leading to optimized performance of the HEP and PHC services, as well as increased utilization and continuity of ANC services and improved perinatal outcomes. The successful integration of quality improvement and community engagement strategies highlights the significant benefits of collaborative efforts among communities, health care facilities, and local authorities. By applying quality improvement principles and team problem-solving to address challenges across various levels of the woreda health system, these combined efforts are likely to contribute to improved perinatal outcomes. These experiences underscore the critical need for sustained multi-level collaboration and robust communication channels to achieve substantial improvements in health systems and ultimately ensure better health outcomes for communities.

## Supporting information

**S1 Fig. Domains of NoCs.**
(TIF)

**S2 Fig. Arrangement of PHC facilities in a woreda NoCs.**
(TIF)

**S3 Fig. Ethiopian Health Center Reform Implementation Guidelines implementation score, September 2022 to February 2024.**
(TIF)

**S1 Table. Roles and responsibilities of each level within the NoCs framework.**
(DOCX)

**S1 File. Key informant interview guide.** This is a key informant interview guide we used to interview the participants in our study.
(DOCX)

**S2 File. Service statistics dataset.** This is service statistics data with variables and their values we used for the analysis with the corresponding data dictionary.
(XLS)

## Acknowledgments

We would like to extend our gratitude to the Ministry of Health, woreda health offices, hospital and health center staff, and Health Extension Workers at the project implementation sites for their critical engagement.

## Author Contributions

**Conceptualization:** Gizachew Tadele Tiruneh, Nebreed Fesseha, Biruhtesfa Bekele Shiferaw.

**Data curation:** Gizachew Tadele Tiruneh, Temesgen Ayehu, Tamar Chitashvili, Mesele Damte Argaw, Mikias Teferi, Agumasie Semahegn, Biruk Bogale, Yibeltal Kifle, Chala Tesfaye.

**Formal analysis:** Gizachew Tadele Tiruneh, Mikias Teferi, Biruk Bogale.

**Funding acquisition:** Gizachew Tadele Tiruneh, Nebreed Fesseha, Temesgen Ayehu, Biruhtesfa Bekele Shiferaw, Dessalew Emaway.

**Investigation:** Dessalew Emaway.

**Methodology:** Gizachew Tadele Tiruneh, Nebreed Fesseha, Temesgen Ayehu, Tamar Chitashvili, Biruhtesfa Bekele Shiferaw, Mikias Teferi, Agumasie Semahegn, Biruk Bogale, Yibeltal Kifle, Hillina Tadesse.

**Project administration:** Gizachew Tadele Tiruneh, Nebreed Fesseha, Temesgen Ayehu, Biruhtesfa Bekele Shiferaw, Hillina Tadesse.

**Supervision:** Dessalew Emaway.

**Validation:** Gizachew Tadele Tiruneh, Nebreed Fesseha, Temesgen Ayehu, Tamar Chitashvili, Mesele Damte Argaw, Mikias Teferi, Agumasie Semahegn, Biruk Bogale, Dessalew Emaway.

**Visualization:** Gizachew Tadele Tiruneh, Nebreed Fesseha, Mesele Damte Argaw, Mikias Teferi, Agumasie Semahegn, Biruk Bogale, Yibeltal Kifle, Hillina Tadesse, Chala Tesfaye, Dessalew Emaway.

**Writing – original draft:** Gizachew Tadele Tiruneh, Temesgen Ayehu, Tamar Chitashvili, Biruhtesfa Bekele Shiferaw, Biruk Bogale, Yibeltal Kifle, Hillina Tadesse, Chala Tesfaye, Dessalew Emaway.

**Writing – review & editing:** Gizachew Tadele Tiruneh, Nebreed Fesseha, Temesgen Ayehu, Tamar Chitashvili, Mesele Damte Argaw, Biruhtesfa Bekele Shiferaw, Mikias Teferi, Agumasie Semahegn, Biruk Bogale, Yibeltal Kifle, Hillina Tadesse, Chala Tesfaye, Dessalew Emaway.

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
