## [Decision Letter · Decision Letter 0]

9 Oct 2024

PONE-D-24-34411Networks of Care for Optimizing Community Health Programs in Ethiopia: Enhancing Linkages and Coordination within Primary Health CarePLOS ONE

Dear Dr. Tiruneh,

Thank you for submitting your manuscript to PLOS ONE. After careful consideration, we feel that it has merit but does not fully meet PLOS ONE’s publication criteria as it currently stands. Therefore, we invite you to submit a revised version of the manuscript that addresses the points raised during the review process.

**ACADEMIC EDITOR:**In order to make the article more compliant with scientific criteria, I suggest eliminating the italicized citations in the text of subjects who, moreover, are not clearly identifiable. Alternatively, they must be reported according to rigorous and standardized criteria.

We look forward to receiving your revised manuscript.

Kind regards,

Flora De Conto, Ph.D.

Academic Editor

PLOS ONE

Journal Requirements:

“We would like to thank the Bill & Melinda Gates Foundation for their generous funding and guidance throughout the project design and implementation”

“The project is funded by the Bill & Melinda Gates Foundation and implemented through a collaborative consortium of Amref Health Africa and JSI. The funder has no role in the interpretation and implications of the content in this paper. This responsibility lies solely with the authors.”

4. We note that your Data Availability Statement is currently as follows: [All data used for this analysis is included in this report.]

Reviewers' comments:

Reviewer's Responses to Questions

**Comments to the Author**

1. Is the manuscript technically sound, and do the data support the conclusions?

Reviewer #1: Partly

2. Has the statistical analysis been performed appropriately and rigorously? 

Reviewer #1: Yes

3. Have the authors made all data underlying the findings in their manuscript fully available?

Reviewer #1: No

4. Is the manuscript presented in an intelligible fashion and written in standard English?

Reviewer #1: Yes

5. Review Comments to the Author

Reviewer #1: The article is comprehensive and provides detailed insights into the topic, but some sections could be streamlined for clarity.

1. The section detailing the Networks of Care (NOC) is overly long and could benefit from conciseness. Summarizing key elements of NOC—such as its role in improving coordination, resource sharing, and maternal care outcomes—would make the manuscript more accessible without losing important information.

2. The mention of the RE-AIM framework feels underdeveloped. If the authors are not applying it fully by reporting on key dimensions like reach, adoption, implementation, and maintenance, it may be better to omit the framework from the methodology or expand its use and provide relevant data for each component.

3. Fidelity should be clearly defined. Currently, it is discussed without specifying what components of the intervention were monitored for adherence, or how deviations were measured. A more structured discussion on how fidelity was evaluated would improve the rigor of the paper.

4. A summary table showing key indicators and the corresponding data collection methods would be helpful for readers. This would clarify how the study measured outcomes and tracked progress over time.

5. The key informant interview methodology was very briefly mentioned ("Qualitative data from key informant interviews were transcribed, coded, and thematically analyzed using Atlas.ti"). I suggest following a standard checklist, such as COREQ, to provide more detailed information on qualitative data collection, including sampling, interview guides, and the coding process.

6. PLOS authors have the option to publish the peer review history of their article (what does this mean?). If published, this will include your full peer review and any attached files.

Reviewer #1: No

---

## [Author Response · Author response to Decision Letter 0]

25 Oct 2024

A point-by-point response to reviews

Dear Editor,

We, the authors, would like to express our gratitude to the reviewers and editors of PLOS ONE Journal for their valuable comments, which we believe have significantly strengthened our paper. Below, we provide point-by-point responses to each comments and suggestions. We have also reviewed the manuscript to ensure that it conforms to the journal's style.

Journal Requirements:

Response: We reviewed to ensure that this version of the manuscript conforms to the journal style.

Response: Thank you. The ethics statement has been revised to include details on consent. The following sentences are now part of the revised version (page 12 lines 214-220): 

“For the retrospective service statistics from medical records, data were collected in monthly aggregates and fully anonymized prior to access. The IRB waived the requirement for client informed consent; however, we obtained consent from facility directors. Informed written consent was also secured from key informants for qualitative interviews. All information obtained from informants was de-identified before analysis, and we maintained strict confidentiality throughout all stages of the study. “

“We would like to thank the Bill & Melinda Gates Foundation for their generous funding and guidance throughout the project design and implementation”

“The project is funded by the Bill & Melinda Gates Foundation and implemented through a collaborative consortium of Amref Health Africa and JSI. The funder has no role in the interpretation and implications of the content in this paper. This responsibility lies solely with the authors.”

Response: Thanks. We have removed the funding-related text from the manuscript. The funding statement contains all necessary declarations and does not require any updates.

4. We note that your Data Availability Statement is currently as follows: [All data used for this analysis is included in this report.]

Response: The qualitative KII guide and the data used for this analysis have been included as Supplementary Information in S4 and S6, respectively.

5. We note that you have included the phrase “data not shown” in your manuscript. Unfortunately, this does not meet our data sharing requirements. PLOS does not permit references to inaccessible data. We require that authors provide all relevant data within the paper, Supporting Information files, or in an acceptable, public repository. Please add a citation to support this phrase or upload the data that corresponds with these findings to a stable repository (such as Fig share or Dryad) and provide and URLs, DOIs, or accession numbers that may be used to access these data. Or, if the data are not a core part of the research being presented in your study, we ask that you remove the phrase that refers to these data.

Response: Thanks. Figure 5 is provided instead (See page 21, lines 408-409)

Review Comments to the Author

Reviewer #1: The article is comprehensive and provides detailed insights into the topic, but some sections could be streamlined for clarity.

1. The section detailing the Networks of Care (NOC) is overly long and could benefit from conciseness. Summarizing key elements of NOC—such as its role in improving coordination, resource sharing, and maternal care outcomes—would make the manuscript more accessible without losing important information.

Response: Thank you for your valuable comment. In this version, we have significantly trimmed this section without losing key information and added S2 Table as supplementary information.

2. The mention of the RE-AIM framework feels underdeveloped. If the authors are not applying it fully by reporting on key dimensions like reach, adoption, implementation, and maintenance, it may be better to omit the framework from the methodology or expand its use and provide relevant data for each component.

Response: Comment acknowledged. Since we did not report on all dimensions of the RE-AIM framework, we have removed it in this version.

3. Fidelity should be clearly defined. Currently, it is discussed without specifying what components of the intervention were monitored for adherence, or how deviations were measured. A more structured discussion on how fidelity was evaluated would improve the rigor of the paper.

Response: Comment acknowledged. In the measurement section, we have now defined fidelity and outlined how we measured it, see Table 1 (page 10, line 188)

4. A summary table showing key indicators and the corresponding data collection methods would be helpful for readers. This would clarify how the study measured outcomes and tracked progress over time.

Response: Comment well taken and Table 1 presents the key indicators measured, along with their definitions and data sources (page 10, line 183)

5. The key informant interview methodology was very briefly mentioned ("Qualitative data from key informant interviews were transcribed, coded, and thematically analyzed using Atlas.ti"). I suggest following a standard checklist, such as COREQ, to provide more detailed information on qualitative data collection, including sampling, interview guides, and the coding process.

Response: Details of the KII data collection, the tool used (included as S4), and the analysis are now included. 

The following paragraph is included in the data collection section (See page 9, lines 177-184). 

“Qualitative data were collected in March 2024 by the project M&E and program managers using an interview guide with open-ended and probing questions to gather insights from program implementers. The main topics of discussion included the differences between NOC and existing linkages, participatory and contextualized design, roles and responsibilities of network members, care coordination and communication, operational impact on PHC and HEP, collaboration and oversight, adherence to standards and quality improvement, monitoring and evaluation mechanisms, support systems and resource sharing, barriers and facilitators, unintended consequences, and areas for improvement (S6 File). Interviews were conducted at convenient locations within the participants' work environment.”

The Analysis section is also revised as follows (see page 12, lines 202-207); 

“All interviews were audio-taped with the consent of the study participants and the records were transcribed verbatim by the project M&E team. The qualitative data from KIIs were transcribed, coded, and thematically analyzed using Atlas.ti. Priori codes were generated from the interview guide. Reports of quotations, themes, and patterns of interrelationships between the themes were identified. and quantitative data were integrated during both the presentation and interpretation stages.”

---

## [Decision Letter · Decision Letter 1]

11 Nov 2024

PONE-D-24-34411R1Networks of Care for Optimizing Primary Health Care Service Delivery in Ethiopia: Enhancing Relational Linkages and Care CoordinationPLOS ONE

Dear Dr. Tiruneh,

Thank you for submitting your manuscript to PLOS ONE. After careful consideration, we feel that it has merit but does not fully meet PLOS ONE’s publication criteria as it currently stands. Therefore, we invite you to submit a revised version of the manuscript that addresses the points raised during the review process.

**The manuscript has been implemented compared to the first version. However, you must complete the review in accordance with the guidance provided by the reviewers. Furthermore, it is necessary to explain the changes that have occurred in the list of authors, since a new author has been added and to certify that all the authors of the manuscript are aware of and in agreement with them, in compliance with the policy of the Journal PLOS ONE.**

We look forward to receiving your revised manuscript.

Kind regards,

Flora De Conto, Ph.D.

Academic Editor

PLOS ONE

**Journal Requirements:**

Reviewers' comments:

Reviewer's Responses to Questions

**Comments to the Author**

1. If the authors have adequately addressed your comments raised in a previous round of review and you feel that this manuscript is now acceptable for publication, you may indicate that here to bypass the “Comments to the Author” section, enter your conflict of interest statement in the “Confidential to Editor” section, and submit your "Accept" recommendation.

Reviewer #1: (No Response)

2. Is the manuscript technically sound, and do the data support the conclusions?

Reviewer #1: Partly

3. Has the statistical analysis been performed appropriately and rigorously? 

Reviewer #1: No

4. Have the authors made all data underlying the findings in their manuscript fully available?

Reviewer #1: No

5. Is the manuscript presented in an intelligible fashion and written in standard English?

Reviewer #1: Yes

6. Review Comments to the Author

**Reviewer #1: **Thank you for the revision. The manuscript has improved in many ways. However, I have included my responses and a few additional comments below (in purple).

Overall comments

Previous responses and comments

1. Thank you for the revision. Table S2 provides an adequate overview of the framework.

2. "The mention of the RE-AIM framework feels underdeveloped. If the authors are not applying it fully by reporting on key dimensions like reach, adoption, implementation, and maintenance, it may be better to omit the framework from the methodology or expand its use and provide relevant data for each component.

Response: Comment acknowledged. Since we did not report on all dimensions of the RE-AIM framework, we have removed it in this version."

Further comments:

In my opinion, if the authors do not expand on their use of the RE-AIM framework (specifying which components guided which indicators and acknowledging any components omitted due to limitations), they should consider mentioning other frameworks that may better suit the manuscript. Otherwise, readers may be confused about how each indicator was selected for analysis.

3. Fidelity should be clearly defined. Currently, it is discussed without specifying what

components of the intervention were monitored for adherence, or how deviations were

measured. A more structured discussion on how fidelity was evaluated would improve

the rigor of the paper.

Response: Comment acknowledged. In the measurement section, we have now

defined fidelity and outlined how we measured it, see Table 1 (page 10, line 188)

Further comments:

Thank you for defining fidelity. However, in line with the previous comment, the presentation of other indicators is confusing, as they don’t seem to clearly adhere to any specific framework.

For example, in the results section of the main text, the authors briefly mention fidelity with the statement: “Fidelity: A sizable progress in NoCs implementation fidelity was achieved.” Then, under the Fidelity subsection, functionality is reported instead: “The NoCs functionality score improved from 56% to 89% in agrarian settings and from 38% to 60% in pastoral settings (Fig 2). Improvements included enhanced coordination, standardized referral practices, and increased resource sharing among facilities. Coaching and mentorship strengthened at primary care levels, improving efficiency, access, and quality of care.”

This is confusing because the authors previously defined functionality as a separate indicator. Following a specific implementation evaluation framework (such as the RE-AIM framework or the MRC Process Evaluation Framework, https://pubmed.ncbi.nlm.nih.gov/25791983/) might clarify the structure and interpretation.

4. “A summary table showing key indicators and the corresponding data collection

methods would be helpful for readers. This would clarify how the study measured

outcomes and tracked progress over time.

Response: Comment well taken and Table 1 presents the key indicators measured,

along with their definitions and data sources (page 10, line 183)”.

Further comments: See comments above.

Additional comments

1. Please avoid overusing abbreviations, as it can make the text difficult to understand. For example, "QI" should be written in full.

2. Results (abstract)

“Results: The NoCs approach has significantly strengthened both administrative and technical connections, while enhancing relational linkages, multi-level collaboration, and bidirectional communication. This has fostered trust, improved care coordination, boosted primary health care performance, and increased health service utilization within woreda health systems. The Interrupted time series analysis indicated that the rate of ANC 8+ visits was 29.8% per month higher than expected without the NoCs strategy (Coef: 2.39; p-value < 0.01) and an 18.4% increase in obstetric complications managed (Coef: 1.71; p-value 36 = 0.050), with a 43% overall increase. Perinatal mortality decreased by 34%, from 31.3 to 20.1 per 1,000 births [t-test: 2.12; p-value: 0.040)].”

Could you clarify the definitions and distinctions between “administrative connections,” “technical connections,” “relational linkages,” “multi-level collaboration,” and “bidirectional communication”?

The results section in the main text reports different indicators than these terms. Could you explain the reason for this discrepancy? Furthermore, why does the abstract use a different format, such as “Implementation fidelity, challenges, and adaptations,” “Enhanced bidirectional linkages,” “Optimized HEP and improved performance and efficiency of the PHC system,” and “Improved utilization, quality, and continuity of care”?

3. Discussions

“…Previous studies underscore that involving communities in these networks and quality improvement efforts significantly boosts maternal health program effectiveness. By engaging community members, local leaders, and stakeholders in service design, implementation, and evaluation, programs become more accessible, culturally sensitive, and widely utilized. Initiatives like participatory learning groups, community health committees, and training for health workers increase antenatal visits, skilled birth attendance, and postnatal care, ultimately lowering maternal and neonatal mortality rates [22-24]. These engaged networks build trust between health care providers and communities, ensuring interventions meet local needs sustainably.”

This section would benefit from additional references. For example, the statement “…Previous studies underscore that involving communities in these networks and quality improvement efforts significantly boosts maternal health program effectiveness” should be followed by citations that demonstrate the effectiveness of “involving communities in these networks…”

Additionally, the discussion section appears to emphasize the importance of the NOC framework without comparing other aspects of the research, such as fidelity, functionality, or the evaluation methodology, with existing literature. Please consider including these comparisons.

4. Please provide the study’s limitations.

5. Please provide necessary information about the shared data in Supplementary #6.csv (e.g., brief descriptions, data dictionary).

7. PLOS authors have the option to publish the peer review history of their article (what does this mean?). If published, this will include your full peer review and any attached files.

Reviewer #1: No

---

## [Author Response · Author response to Decision Letter 1]

15 Nov 2024

A point-by-point response to reviews

Dear Editor,

We, the authors, would like to express our gratitude to the reviewers and editors of PLOS ONE Journal for their valuable comments, which we believe have significantly strengthened our paper. Below, we provide point-by-point responses to each comment and suggestions in red font. We have also reviewed the manuscript to ensure that it conforms to the journal's style.

Editor comment

The manuscript has been implemented compared to the first version. However, you must complete the review in accordance with the guidance provided by the reviewers. Furthermore, it is necessary to explain the changes that have occurred in the list of authors, since a new author has been added and to certify that all the authors of the manuscript are aware of and in agreement with them, in compliance with the policy of the Journal PLOS ONE.

Response: Thank you. We have carefully addressed all the feedback provided by the reviewer in this version. Additionally, including a new author has been communicated to all contributors. We inadvertently omitted his name from the author list in the initial submission. However, he contributed substantially to the manuscript, including project design, data acquisition, conceptualization, and review. 

Journal Requirements:

Response: The reference lists have been reviewed, and minor corrections have been made to the format

Reviewers' comments:

Reviewer #1: Thank you for the revision. The manuscript has improved in many ways. However, I have included my responses and a few additional comments below (in purple).

Response: Thank you for the valuable comments, which have been essential for enhancing the scientific rigor of this manuscript. We have carefully addressed all feedback in this version. 

Overall comments

Previous responses and comments

1. Thank you for the revision. Table S2 provides an adequate overview of the framework.

Response: Thank you. 

2. "The mention of the RE-AIM framework feels underdeveloped. If the authors are not applying it fully by reporting on key dimensions like reach, adoption, implementation, and maintenance, it may be better to omit the framework from the methodology or expand its use and provide relevant data for each component.

Response: Comment acknowledged. Since we did not report on all dimensions of the RE-AIM framework, we have removed it in this version."

Further comments:

In my opinion, if the authors do not expand on their use of the RE-AIM framework (specifying which components guided which indicators and acknowledging any components omitted due to limitations), they should consider mentioning other frameworks that may better suit the manuscript. Otherwise, readers may be confused about how each indicator was selected for analysis.

Response: Thank you again for the valid and valuable feedback. In this version, Table 1 has been substantially revised, with indicators mapped to the RE-AIM outcomes. Some outcome measures, such as Maintenance (sustainability), are not adequately measured and are addressed in the limitations section of the manuscript. See Pages 8-9, lines 157-159, and Table 1 pages 10-11, lines 193-197 as well as page 14, lines 230-234 on Revised Manuscript with Track Changes. 

3. Fidelity should be clearly defined. Currently, it is discussed without specifying what

components of the intervention were monitored for adherence, or how deviations were

measured. A more structured discussion on how fidelity was evaluated would improve

the rigor of the paper.

Response: Comment acknowledged. In the measurement section, we have now

defined fidelity and outlined how we measured it, see Table 1 (page 10, line 188)

Further comments:

Thank you for defining fidelity. However, in line with the previous comment, the presentation of other indicators is confusing, as they don’t seem to clearly adhere to any specific framework.

For example, in the results section of the main text, the authors briefly mention fidelity with the statement: “Fidelity: A sizable progress in NoCs implementation fidelity was achieved.” Then, under the Fidelity subsection, functionality is reported instead: “The NoCs functionality score improved from 56% to 89% in agrarian settings and from 38% to 60% in pastoral settings (Fig 2). Improvements included enhanced coordination, standardized referral practices, and increased resource sharing among facilities. Coaching and mentorship strengthened at primary care levels, improving efficiency, access, and quality of care.”

This is confusing because the authors previously defined functionality as a separate indicator. Following a specific implementation evaluation framework (such as the RE-AIM framework or the MRC Process Evaluation Framework, https://pubmed.ncbi.nlm.nih.gov/25791983/) might clarify the structure and interpretation.

Response: As described above, in this version, fidelity is measured and defined by NoC functionality and maturity metrics, as well as by contextual challenges and adaptations in alignment with the RE-AIM framework. Table 1 pages 10-11, lines 193-197 on Revised Manuscript with Track Changes

4. “A summary table showing key indicators and the corresponding data collection

methods would be helpful for readers. This would clarify how the study measured

outcomes and tracked progress over time.

Response: Comment well taken and Table 1 presents the key indicators measured,

along with their definitions and data sources (page 10, line 183)”.

Further comments: See comments above.

Response: The indicators measured are presented in the revised Table 1, along with their definitions and data sources, in alignment with the RE-AIM framework. Table 1 pages 10-11, lines 193-197 on Revised Manuscript with Track Changes

Additional comments

1. Please avoid overusing abbreviations, as it can make the text difficult to understand. For example, "QI" should be written in full.

Response: Thank you for bringing this to our attention. In this version, we have removed abbreviations that are not frequently used in the text and written them out in full, including terms like QI, PDSA, FRAME-IS, ERIC, WDA, MOU, and EHCRIG.

2. Results (abstract)

“Results: The NoCs approach has significantly strengthened both administrative and technical connections, while enhancing relational linkages, multi-level collaboration, and bidirectional communication. This has fostered trust, improved care coordination, boosted primary health care performance, and increased health service utilization within woreda health systems. The Interrupted time series analysis indicated that the rate of ANC 8+ visits was 29.8% per month higher than expected without the NoCs strategy (Coef: 2.39; p-value < 0.01) and an 18.4% increase in obstetric complications managed (Coef: 1.71; p-value 36 = 0.050), with a 43% overall increase. Perinatal mortality decreased by 34%, from 31.3 to 20.1 per 1,000 births [t-test: 2.12; p-value: 0.040)].”

Could you clarify the definitions and distinctions between “administrative connections,” “technical connections,” “relational linkages,” “multi-level collaboration,” and “bidirectional communication”?

The results section in the main text reports different indicators than these terms. Could you explain the reason for this discrepancy? Furthermore, why does the abstract use a different format, such as “Implementation fidelity, challenges, and adaptations,” “Enhanced bidirectional linkages,” “Optimized HEP and improved performance and efficiency of the PHC system,” and “Improved utilization, quality, and continuity of care”?

Response: Thank you for the comments. We have revised the abstract (Results) section to provide greater clarity for readers as shown below. 

“The NoCs approach has significantly enhanced relational linkages between primary health care facilities and health care providers, fostering stronger collaboration and communication. This has fostered trust, improved care coordination, optimized primary health care performance, and increased health service utilization within woreda health systems. The Interrupted time series analysis indicated that the rate of ANC 8+ visits was 29.8% per month higher than expected without the NoCs strategy (Coef: 2.39; p-value < 0.01) and an 18.4% increase in obstetric complications managed (Coef: 1.71; p-value = 0.050), with a 43% overall increase. Perinatal mortality decreased by 34%, from 31.3 to 20.1 per 1,000 births [t-test: 2.12; p-value: 0.040)].” See Pages 2-3, lines 31-40 on Revised Manuscript with Track Changes. 

3. Discussions

“…Previous studies underscore that involving communities in these networks and quality improvement efforts significantly boosts maternal health program effectiveness. By engaging community members, local leaders, and stakeholders in service design, implementation, and evaluation, programs become more accessible, culturally sensitive, and widely utilized. Initiatives like participatory learning groups, community health committees, and training for health workers increase antenatal visits, skilled birth attendance, and postnatal care, ultimately lowering maternal and neonatal mortality rates [22-24]. These engaged networks build trust between health care providers and communities, ensuring interventions meet local needs sustainably.”

This section would benefit from additional references. For example, the statement “…Previous studies underscore that involving communities in these networks and quality improvement efforts significantly boosts maternal health program effectiveness” should be followed by citations that demonstrate the effectiveness of “involving communities in these networks…”

Response: Thank you 3 references are now included as suggested. Page 25, line 458 on Revised Manuscript with Track Changes

Additionally, the discussion section appears to emphasize the importance of the NOC framework without comparing other aspects of the research, such as fidelity, functionality, or the evaluation methodology, with existing literature. Please consider including these comparisons.

Response: Thank you for your valuable comments. In this version, we compared the use of implementation science research methodologies, bridging the knowledge gap left by previous studies focused on clinical experiences and case study descriptions. This study explored the mechanisms shaping networks between PHC facilities and communities. Additionally, we discussed how fidelity and adaptation influence implementation outcomes and addressed the limitations caused by inadequate fidelity and program implementation, particularly in assessing RE-AIM outcomes like Maintenance (sustainability). Page 26 lines 493-495; Page 27, lines 497-498; and lines 507-513 on Revised Manuscript with Track Changes

4. Please provide the study’s limitations.

Response: Thank you for the valid comment. The following paragraph is included in this version. 

"Adapting programs to local contexts is crucial for ensuring strategies are both practical and effective. High-fidelity implementation improves PHC performance, health outcomes, and sustainability. However, due to the study’s limited duration, certain RE-AIM measures, such as sustainability (Maintenance), were not fully assessed, limiting the evaluation of NoCs’ maturity and integration into health system processes. Additionally, the absence of a comparison group may weaken the evidence from the time-series analysis, and the functionality and maturity matrix could be influenced by assessor bias and implementation variability." Page 27, lines 507-513 on Revised Manuscript with Track Changes

5. Please provide necessary information about the shared data in Supplementary #6.csv (e.g., brief descriptions, data dictionary).

Response: Apologies for overlooking the data dictionary. In this version, a data dictionary and brief description have been included as separate sheets labeled 'Data Dictionary' and 'Description.'

---

## [Editor Report · Decision Letter 2]

18 Nov 2024

Networks of Care for Optimizing Primary Health Care Service Delivery in Ethiopia: Enhancing Relational Linkages and Care Coordination

PONE-D-24-34411R2

Dear Dr. Tiruneh,

We’re pleased to inform you that your manuscript has been judged scientifically suitable for publication and will be formally accepted for publication once it meets all outstanding technical requirements.

Kind regards,

Flora De Conto, Ph.D.

Academic Editor

PLOS ONE
---

## [Editor Report · Acceptance letter]

13 Dec 2024

PONE-D-24-34411R2 

PLOS ONE

Dear Dr. Tiruneh, 

I'm pleased to inform you that your manuscript has been deemed suitable for publication in PLOS ONE. Congratulations! Your manuscript is now being handed over to our production team.

Kind regards, 

on behalf of

Prof. Flora De Conto 

Academic Editor

PLOS ONE